# Excessive boredom among adolescents: A comparison between low and high achievers

**Manuel M. Schwartze** [ORCID]**¹, Anne C. Frenzel¹, Thomas Goetz²\*, Anton K. G. Marx¹, Corinna Reck¹, Reinhard Pekrun¹,³,⁴, Daniel Fiedler¹**

**1** Department of Psychology, Ludwig-Maximilians-Universität München, Munich, Germany, **2** Department of Developmental and Educational Psychology, Faculty of Psychology, University of Vienna, Vienna, Austria, **3** Department of Psychology, University of Essex, Colchester, United Kingdom, **4** Institute for Positive Psychology and Education, Australian Catholic University, Sydney, Australia

\* thomas.goetz@univie.ac.at

**Data Availability Statement:** All data files are available from the Open Science Framework database (https://osf.io/zypae).

## Abstract

Existing research shows that high achievement boredom is correlated with a range of undesirable behavioral and personality variables and that the main antecedents of boredom are being over- or under-challenged. However, merely knowing that students are highly bored, without taking their achievement level into account, might be insufficient for drawing conclusions about students' behavior and personality. We, therefore, investigated if low- vs. high-achieving students who experience strong mathematics boredom show different behaviors and personality traits. The sample consisted of 1,404 German secondary school students (fifth to 10th grade, mean age 12.83 years, 52% female). We used self-report instruments to assess boredom in mathematics, behavioral (social and emotional problems, positive/negative affect, cognitive reappraisal, and expressive suppression), and personality variables (neuroticism and conscientiousness). In comparing highly bored students (more than one $SD$ above $M$, $n = 258$) who were low vs. high achievers (as indicated by the math grade, $n = 125 / n = 119$), results showed that there were no mean level differences across those groups for all variables. In conclusion, our results suggest that high boredom can occur in both low- and high-achieving students and that bored low- and high-achievers show similar behaviors and personality profiles.

## Introduction

Boredom is one of the most commonly experienced emotions in educational settings [1,2]. Adolescents report being bored 30–40% of the time in school [3,4], but also in their spare time [5]. Highly bored students were shown to avoid schoolwork [6], to have attention problems, and reduced effort, self-regulation, and motivation [7–9]. They were also shown to use less effective learning strategies [2,9]. As a consequence, there is consistent evidence that boredom correlates negatively with academic achievement [9–17]. More generally, high boredom among adolescents has been associated with numerous serious problems like dropping out of school [4,18] or juvenile delinquency [19,20]. An important and well-documented characteristic of boredom is that it can be triggered by both over- and under-challenge [21]. However, it is unclear whether boredom is similarly severe when students are bored due to over-challenge

**Funding:** This research was supported by a grant awarded to the second, fifth and sixth author by the Deutsche Forschungsgemeinschaft (FR 2642/8-1 and RE 2249/4-1). Open access funding provided by University of Vienna. The funders had no role in study design, data collection and analysis, decision to publish, or preparation of the manuscript.

**Competing interests:** The authors have declared that no competing interests exist.

and when they are bored due to under-challenge. In other words: Are undesirable correlates of boredom worse in the case of over-challenge, and may under-challenged students not suffer as much? Or is it the excessive boredom per se that covaries with problematic behavior and personality? To address this question, we systematically compared students who are highly bored and low-achieving, that is, likely over-challenged, and highly bored yet high-achieving, that is, likely under-challenged, in the subject of mathematics. The present study thus seeks to enrich the literature by enhancing our understanding of achievement boredom. Specifically, we add further knowledge about a potential differentiation between boredom due to being over- vs. under-challenged and offer practical implications for teachers, students, and parents.

## Boredom as an unpleasant emotion with undesirable correlates

Boredom, most generally, is described as an unpleasant and distressing experience [22]. There are two widely used scales to measure general trait boredom: The Boredom Proneness Scale [BPS; 23] and the Boredom Susceptibility Scale [BSS; 24]. Research on the BPS has revealed that boredom proneness has multiple undesirable correlates, including alexithymia [25], alienation [26], anger and aggression [27–29], impulsiveness [28,30–32], loneliness [23], narcissism [33], negative affect [34], neuroticism [28,35,36] procrastination [37,38], and unsociability [31]. In turn, low levels of boredom proneness have been shown to be linked with higher levels of conscientiousness, openness to experience [6], and life satisfaction [23]. High scores on the BSS have been reported to be associated with higher levels of motor impulsivity, sensitivity to reward, gambling, alcohol, and smoking [36,39]. Going beyond such general, context-transcending findings, the present study specifically addresses boredom at school, and even more specifically, student experiences of boredom in the subject of mathematics. We thus assess boredom as a trait construct in a domain-specific way.

While mathematics boredom has been studied in several recent studies addressing, for example, the control- and value-appraisal antecedents of mathematics boredom [9,40], or boredom-achievement links [41], no study to date seems to have explored whether such domain-specific boredom is also linked with person-level behavioral and personality variables. In other words, it remains open to question if those students who report to experience intense boredom in mathematics only show undesirable levels of structs related to the domain of mathematics (e.g., poor study habits), or if they also show problematic behavior patterns beyond this context (e.g., lower sociability). In line with Bronfenbrenner's [42] ecological systems theory, we suggest that domain-specific boredom and more general behavioral and personal variables inevitably interact with each other. Thus, the first aim of this study was to replicate prior correlational findings as demonstrated using more general instruments for the assessment of boredom in the subject of mathematics.

## Boredom due to being over- vs. under-challenged

The idea of boredom being caused by under-challenge has already been brought forward by Csikszentmihalyi in 1975 [43]. In this work, he argued that boredom supposedly arises in situations in which someone's competencies are higher than the situational opportunities or, in other words, in situations that are under-challenging. However, boredom can also be prompted when task demands are too high and cannot be interpreted in a meaningful way, implying over-challenge [8]. Integrating across both perspectives, Pekrun's [44,45] control-value theory of achievement emotions proposes that boredom should be linked with either low or high control. In other words, according to this theory, students should experience boredom when they appraise that success is either quite easily or only barely attainable for them [40]. This implies that both low and high achievers may experience high levels of boredom. Over

the past years, these theoretical propositions have been addressed by a large body of empirical research which has consistently demonstrated that boredom is, indeed, experienced in both over- and under-challenging situations [21,46–50].

Despite this compelling evidence on the meaning of differentiating between boredom due to being over- vs. under-challenged, what still seems open to question is whether experiencing intense boredom is similarly severe when students are low-achieving and thus likely over-challenged, or when students are high-achieving and thus likely under-challenged. On the one hand, the undesirable correlates of boredom may arise only for poorly performing students, while high performing students may not suffer as much from undesirable correlates of boredom. Such reasoning would be supported by the fact that high academic achievement typically is associated with conscientiousness [51,52] and high self-esteem [53]. Those factors could protect against the potential undesirable correlates of boredom. From another perspective, experiencing intense levels of boredom at school may imply undesirable correlates, irrespective of levels of challenge, and scholastic performance. Such reasoning is supported by Kannich's [21] study which showed both being over- or under-challenged resulted in a decrease in career aspirations.

## The present study

The present study addresses a gap in research on achievement boredom by systematically comparing students who are highly bored and low-achieving–thus, likely over-challenged, and highly bored yet high-achieving–thus, likely under-challenged. As potential undesirable correlates, we took into account both behavioral and personality variables. As achievement boredom has been shown to be highly domain-specific [12] and particularly salient in mathematics [49] we decided to focus on this domain. The present study takes a trait perspective [44], proposing that individuals systematically differ in their tendency to experience boredom.

The choice of constructs addressed in the present study was guided by the aim to address the central negative aspects mentioned in the general boredom proneness literature, inasmuch as they seemed relevant in our context. We thus aimed at replicating prior findings on a broad range of correlates of boredom as demonstrated using more general instruments for the assessment of boredom proneness, while assessing boredom specifically with respect to the subject of mathematics. Previous research has shown that boredom is linked with enhanced negative emotions [29], conduct problems [20,27,31], hyperactivity [54], peer problems [26], and lack of prosocial behavior [31]. Therefore, to explore potential undesirable correlates of boredom, we took all subscales of the Strength and Difficulties Questionnaire [SDQ, 55] into account. Furthermore, boredom has been shown to be positively linked with negative affect [56], expressive suppression [57], and neuroticism [36] as well as negatively with positive affect [56], cognitive reappraisal [57], and conscientiousness [6]. We therefore additionally considered general affect as measured with the Positive and Negative Affect Schedule [PANAS; 58], cognitive reappraisal and expressive suppression as measured with the Emotion Regulation Questionnaire [ERQ; 59] and finally, neuroticism and conscientiousness as measured with the Big Five Inventory-2 [BFI-2; 60].

Despite the extensive body of research examining achievement boredom in adolescents, it is still open to question whether experiencing intense boredom is similarly severe when students are low-achieving and when they are high-achieving. Therefore, we formulated the following exploratory research question: Do low-achieving students with high boredom systematically differ in their self-reported behaviors and personality traits from high-achieving students with high boredom? We propose that an answer to this question enhances the scientific understanding of achievement boredom and offers practical implications, especially with

respect to potentially dealing differentially with students who are bored due to being over- vs. under-challenged.

## Method

### Sample

The sample consisted of $N = 1.404$ secondary school students from 103 classrooms of 25 schools (52% girls [$n = 731$], 47% boys [$n = 661$], 1% not indicated [$n = 12$]) from the Free State of Bavaria, Germany. Students were from all three tracks of the Bavarian three-track general secondary school system, with 47% ($n = 662$ students) from the upper (Gymnasium), 28% ($n = 390$) the middle (Realschule), and 25% ($n = 349$) the lower track (Mittelschule). This distribution across tracks is equivalent with the Bavarian secondary student statistics, with a slight overrepresentation of Gymnasium student population [61]. The students were in the fifth ($n = 172$), sixth ($n = 197$), seventh ($n = 582$), eighth ($n = 291$), ninth ($n = 134$), and 10th grade ($n = 24$) and were 9 to 17 years old, with a mean age of $M_{age} = 12.83$ years ($SD_{age} = 1.29$). The vast majority of the students (92%, $n = 1.287$) was born in Germany while 18% of them had at least one foreign-born parent ($n_{mother} = 181$, $n_{father} = 177$, $n_{both} = 118$).

The research was approved by Ludwig Maximilian University of Munich's Ethics Review Board of the Faculty of Psychology and Education. Participation in the study was voluntary, written informed consent was obtained from all participants, parents or guardians respectively, and no identifiers that could link individual participants to their results were obtained.

### Measures

The data reported here were assessed as part of a longer questionnaire which in total consisted of ten pages with open-ended and multiple-choice questions. External trained testing personnel brought the questionnaires to the schools and collected them a few weeks later. The questionnaire was filled out at home by the students and collected, inside sealed envelopes, in class by their mathematics teachers.

**Boredom.** Students' class-related, habitual, trait-like boredom in mathematics was accessed using six items of the course-specific boredom scale of the Achievement Emotions Questionnaire—Mathematics [15,AEQ-M, 62]. In the AEQ, students are prompted to *"Please indicate how you feel, typically, during math class"*; a sample item is "I am so bored that I can't stay awake" (see Table 1 for the full set of items used in this study in original German, and their English translation). Students responded using a 5-point Likert scale ranging from 1 (*strongly disagree*) to 5 (*strongly agree*).

**Table 1. Boredom items of the Achievement Emotions Questionnaire—Mathematics (AEQ-M).**

| Items German | Items English translation |
|---|---|
| Ich finde den Unterricht langweilig. | I think the mathematics class is boring. |
| Vor Langeweile schalte ich ab. | I can't concentrate because I am so bored. |
| Vor Langeweile kann ich mich kaum wach halten. | I am so bored that I can't stay awake. |
| Vor Langeweile gehen mir immer wieder Gedanken durch den Kopf, die mit Mathe nichts zu tun haben. | I think about what else I might be doing rather than sitting in this boring class. |
| Ich schaue ständig auf die Uhr, weil die Zeit nicht vergeht. | Because of time drags I frequently look at my watch. |
| Ich werde unruhig, weil ich nur darauf warte, dass die Mathestunde endlich vorüber ist. | I get restless because I can't wait for the class to end. |

Asking students to judge "Please indicate how you feel, typically, during math class."

**Achievement.**   Self-reported math grades from students' last final report card were used as an indicator of achievement. The grades are summative scores based on multiple evaluations over the course of a school year and range from 6 (*poor*) to 1 (*excellent*).

**Emotional and behavioral problems.**   The German version [SDQ-Deu-S; 63] of the one-sided self-report version [see 64] of the Strengths and Difficulties Questionnaire for 11–17 year-olds by Goodman [55] was used to measure emotional and behavioral problems. The items comprised of five subscales of five items each for emotional symptoms (e.g., "I worry a lot"), conduct problems ("I get very angry and often lose my temper"), hyperactivity ("I am restless, I cannot stay still for long"), peer problems ("I would rather be alone than with people of my age"), and prosocial behavior ("I am helpful if someone is hurt, upset or feeling ill"). Students were asked to judge these items on a scale from 1, *not true*, 2, *somewhat true*, to 3, *certainly true*.

**Positive and negative affect.**   The German version by Krohne, Egloff, Kohlmann, and Tausch [65] of the Positive and Negative Affect Schedule [PANAS; 58] was used to determine students' general affective states. This self-report scale consists of 10 positive (e.g., "excited") and 10 negative adjectives (e.g., "upset"). Participants responded on a 5-point Likert scale ranging from 1 (*not at all*) to 5 (*extremely*) to describe their "general emotional state."

**Cognitive reappraisal and expressive suppression.**   The German version of the Emotion Regulation Questionnaire [see 59 for the English version,ERQ; 66] was used to measure the tendency to regulate emotions by cognitive reappraisal or expressive suppression. Participants had to rate four items on cognitive reappraisal (e.g., "When I'm faced with a stressful situation, I make myself think about it in a way that helps me stay calm") and expressive suppression (e.g., "I keep my emotions to myself") on a scale from 1 (*not at all true*) to 7 (*completely true*).

**Conscientiousness and neuroticism.**   We considered two of the big five personality traits which have been reported to be systematically linked with boredom, namely conscientiousness, and neuroticism. While conscientiousness (e.g., "I am someone who is systematic, likes to keep things in order") measures differences in organization, productiveness, and responsibility, neuroticism (e.g., "I am someone who tends to feel depressed, blue") measures differences in the frequency and intensity of negative emotions [67]. We used the German version of the Big Five Inventory-2 for their assessment [see 60 for the English version,BFI-2; 68]. Students were asked to rate 12 items for each construct on a 5-point rating scale ranging from 1 (*strongly disagree*) to 5 (*strongly agree*).

## Data analyses

All analyses were conducted using R 3.6.1 [45]. The full analysis code is available from the Open Science Framework database (https://osf.io/zypae). To assess the internal consistency of the scales, the reliability coefficient Cronbach's alpha (α) was calculated. As outlined in Table 2, AEQ-M boredom, PANAS positive and negative affect, and BFI-2 neuroticism and conscientiousness showed good reliabilities (α between .81 and .86). SDQ hyperactivity and prosocial behavior, ERQ cognitive reappraisal, and expressive suppression showed borderline-acceptable reliabilities, but SDQ conduct and peer problems showed low reliabilities (α between .47 and .53). However, earlier studies also documented comparably low internal consistencies for those SDQ subscales when using student ratings [69]. Therefore, this was not a peculiarity of our sample. To circumvent biased results due to scale unreliability, we chose to model all variables as latent constructs using the Lavaan 0.6–5 package [70] employing the full information likelihood method [FIML; 71] for treating missing data, and the MLR estimator (maximum likelihood estimation with robust (Huber-White) standard errors and a scaled test statistic that is (asymptotically) equal to the Yuan-Bentler test statistic).

**Table 2. Means, standard deviations, and Cronbach's alpha for the study scales.**

| Scale | Construct | M (SD) | Min.—Max. | α |
|---|---|---|---|---|
| AEQ-M | Boredom | 2.39 (0.95) | 1.00–5.00 | .86 |
| SDQ | Emotional symptoms | 1.58 (.48) | 1.00–3.00 | .71 |
| | Conduct problems | 1.36 (.31) | 1.00–3.00 | .47 |
| | Hyperactivity | 1.73 (.45) | 1.00–3.00 | .68 |
| | Peer problems | 1.42 (.34) | 1.00–3.00 | .53 |
| | Prosocial behavior | 2.61 (.36) | 1.00–3.00 | .65 |
| PANAS | Positive affect | 3.53 (.62) | 1.00–5.00 | .80 |
| | Negative affect | 2.00 (.68) | 1.00–4.80 | .84 |
| ERQ | Cognitive reappraisal | 3.93 (1.25) | 1.00–7.00 | .68 |
| | Expressive suppression | 3.70 (1.25) | 1.00–7.00 | .60 |
| BFI-2 | Neuroticism | 2.65 (.66) | 1.00–5.00 | .81 |
| | Conscientiousness | 3.40 (.67) | 1.25–5.00 | .82 |

$1390 \leq n \leq 1404$ due to missing values. α = Cronbach's alpha.

We thus obtained latent correlations between boredom, emotional and behavioral problems, positive and negative affect, cognitive reappraisal, and expressive suppression, as well as neuroticism and conscientiousness based on structural equation modeling (SEM). To identify highly bored students, we obtained latent factor scores for each student for the six items of the AEQ-M boredom scale. In this context, we defined the high boredom group to include all students who scored higher than one standard deviation ($SD = 0.7$) above the standardized sample mean of zero on the AEQ-M boredom scale ($n = 258$). To compare across low- vs. high-achievers among these highly bored students, we used the final math grade of the previous school year as an indicator of achievement in math class. In this analysis, students with missing grades ($n = 14$) were excluded. Grades from 4 to 6 (4 = sufficient, 5 = poor, 6 = insufficient) were coded as 0 = low achievement and grades from 1 to 3 (1 = excellent, 2 = good, 3 = satisfactory) as 1 = high achievement ($M = 3$, $SD = 0.9$, $Mdn = 4$). As a result, there were 125 students in the low achievement group (boredom $M = 3.98$, $SD = .53$), and 119 students in the high achievement group (boredom $M = 3.80$, $SD = .42$). To account for multiple testing, we used the Bonferroni method to adjust the alpha level to 0.005.

## Results

### Preliminary analysis

Table 3 shows the latent correlations between students' mathematics boredom and all other affective and behavioral constructs considered in this study, across the full sample. Boredom correlated significantly with all other constructs assessed. Strong relations were found for conduct problems and hyperactivity ($r$ between .52 and .56), and medium-sized relations were found for emotional symptoms, positive and negative affect, and neuroticism and conscientiousness ($r$ between -.45 and .45). Peer problems and prosocial behavior, as well as cognitive reappraisal and expressive suppression, showed small-sized links with mathematics boredom ($r$ between -.29 and .13). The overall pattern of relationships was consistent with previous studies on boredom proneness in that higher levels of boredom in mathematics class were associated with higher levels of undesired behavioral and personality variables, and lower levels of desirable behavioral and personality variables.

**Table 3. Behavior and personality: Latent correlations with boredom and comparison between bored low and high achievers.**

| | | | | Manifest means | | | | |
| Scale | Construct | Latent correlation with boredom | | Low achievers | High achievers | Comparison of latent means | | |
| | | r | p | M (SD) | M (SD) | β | p | R² |
| SDQ | Emotional symptoms | .42 | < .001 | .90 (.52) | .79 (.53) | -.14 | .067 | .02 |
| | Conduct problems | .52 | < .001 | .61 (.43) | .46 (.35) | -.23 | .005 | .05 |
| | Hyperactivity | .56 | < .001 | 1.07 (.46) | .99 (.49) | -.04 | .555 | < .00 |
| | Peer problems | .25 | < .001 | .56 (.40) | .47 (.35) | -.17 | .091 | .03 |
| | Prosocial behavior | -.29 | < .001 | 1.48 (.42) | 1.50 (.40) | .01 | .874 | < .00 |
| PANAS | Positive affect | -.45 | < .001 | 3.16 (.71) | 3.40 (.62) | .20 | .009 | .04 |
| | Negative affect | .40 | < .001 | 2.40 (.67) | 2.28 (.67) | -.12 | .110 | .01 |
| ERQ | Cognitive reappraisal | -.12 | .002 | 3.69 (1.35) | 3.85 (1.16) | .12 | .162 | .01 |
| | Expressive suppression | .13 | .001 | 4.09 (1.32) | 3.63 (1.26) | -.19 | .041 | .04 |
| BFI-2 | Neuroticism | .45 | < .001 | 3.06 (.71) | 2.91 (.64) | -.11 | .199 | .01 |
| | Conscientiousness | -.44 | < .001 | 3.02 (.67) | 3.06 (.65) | -.02 | .774 | < .00 |

*Bonferroni adjusted p-value < .005. $R^2$ = coefficient of determination.*

## Group differences between low and high performers

Before comparing latent mean differences between low- and high-achieving students, we tested for measurement invariance of each of the latent constructs addressed in this study, using the SemTools 0.5–2 package [72]. This was to make sure that the latent scores used in the analysis were comparable across both groups. We sequentially tested for equivalence of model form (configural), equivalence of factor loadings (metric), and equivalence of item intercepts or thresholds [scalar; 73]. For comparing latent means across groups, scalar invariance is necessary [74]. We refrained from additionally testing for residual invariance, which is nugatory to the interpretation of latent mean differences [74]. As can be seen from S1 Table, scalar factorial invariance could indeed be accepted for all constructs except SDQ hyperactivity and peer problems. While hyperactivity showed metric invariance, peer problems only showed configural invariance, implying considerably different item functioning of those items for the low- as opposed to high-achieving bored students.

To investigate differences in behavioral and personality variables of highly bored students who are performing poorly vs. well in mathematics, we regressed the dichotomous variable achievement in mathematics (low vs. high) on all other constructs considered in this study, modeled as latent variables. The results (Table 3) revealed no group differences for any of the constructs. It is worth noting that those results proved to be fully robust when entering school type as dummy-coded control variables. In interpreting these results, differential item functioning for hyperactivity and peer problems must be taken into account.

## Discussion

In the present study, we aimed to systematically compare students who are highly bored and low-achieving, i.e., likely over-challenged, with students who are highly bored and high-achievement, i.e., likely over-challenged. We argued that it remains open to question whether experiencing intense boredom is associated with similarly severe levels of undesirable correlates when students are low- vs. high-achieving. To this end, within the group of highly bored students in our sample, we compared across low-achieving and thus likely over-challenged, and high-achieving and thus likely under-challenged students.

As a preliminary analysis step, we examined correlates of students' boredom in the context of mathematics, following up on previous research which has consistently reported that boredom has multiple undesirable correlates. Our results fully replicated earlier-reported patterns of relationships with undesirable boredom correlates. Specifically, we found again that student-reported experiences of boredom during mathematics classes is positively correlated with emotional and behavioral problems, negative affectivity, the use of expressive suppression to regulate emotions, and neuroticism. In contrast, students' mathematics boredom proved to be negatively correlated with levels of prosocial behavior, positive affectivity, cognitive reappraisal, and conscientiousness.

Moreover, and most importantly, our results suggest that high boredom is associated with similar levels of problematic correlates in low- and high-achieving students. The two groups did not significantly differ in emotional symptoms, conduct problems, hyperactivity, peer problems, prosocial behavior, positive and negative affect, neuroticism, cognitive reappraisal and expressive suppression, neuroticism, and conscientiousness. In line with Pekrun's [44,45] control-value theory of achievement emotions which posits that boredom can occur either when control is particularly high, or when it is particularly low, we find that both over- and under-challenge can lead to high boredom. Furthermore, irrespective of student's performance, and hence irrespective of their subjective control in a certain domain, our study demonstrates that high boredom itself is associated with many of these problems. In sum, we propose that one important implication from our findings is that boredom is boredom–irrespective of its antecedents.

## Limitations, suggestions for future research, and implications

By showing that bored low- and high-achievers show similar patterns in behavioral and personality variables, this study addresses a gap in boredom research and contributes to a better understanding of achievement boredom. However, the following limitations should be taken into account when interpreting our results and could be considered as directions for future research.

First of all, the present study relies on the reasoning that the combination of high boredom with good grades in mathematics implies that those students tend to be bored due to being under-challenged, while the combination of high boredom with poor grades implies that they tend to be over-challenged. It is important to note that this is an assumption, and the classification as over- vs. under-challenge may not have been fully valid for each individual student in the two groups. However, we deliberately chose to assess domain-specific boredom and domain-specific achievement separately, to first identify students with very high boredom, and then classify boredom as likely being due to over- vs. under-challenge based on students' achievement. While this indirect approach to assess over- and under-challenge may be a point of debate, we also deem more direct self-report assessments (e.g., 'I am bored because it's too easy') as psychometrically problematic. Items combining reports of boredom with attributions of boredom are double-barreled and thus ambiguous–it is unclear if students who endorse those items do so because they are bored, or because they find the material easy vs. hard, or because they attribute boredom to over- or under-challenge.

Moreover, our study was conducted in math class at secondary schools in Germany. To generalize our findings, future research should consider problematic correlates of intense boredom in high- and low-achievers in other relevant contexts like elementary schools, universities, or the workplace; in domains other than mathematics; and in other cultures.

With almost 20% ($n$ = 256) of the students in our sample indicated to be severely bored in math class, this study suggests again that no student should be left alone to endure the

"torments of boredom" [75]. Given that students almost exclusively use avoidance-oriented coping strategies to deal with their boredom [76], boredom should be openly discussed in class, and more promising coping strategies such as cognitive- and behavioral-approach strategies should be addressed [77].

One of the most reported reasons for boredom is low-quality instructional design [78]. An adaptive and individualized learning environment might, therefore, contribute to preventing boredom due to being both over- or under-challenged. Most importantly, teachers, parents, and students should be aware that boredom in school needs to be taken seriously. Boredom can indicate severe problems not just in the sense of a student being lazy, too bright, over-challenged, or under-challenged, but can constitute a debilitating personality trait.

## Supporting information

**S1 Table. Chi-Squared difference test for the nested model comparison.** Total $N$ = 244; group 1 $n$ = 125; group 2 $n$ = 119. M1: Configural invariance. M2: Metric invariance. M3: Scalar invariance. [**] $p \leq .01$.
(PDF)

## Author Contributions

**Conceptualization:** Manuel M. Schwartze, Anne C. Frenzel.

**Data curation:** Manuel M. Schwartze.

**Formal analysis:** Manuel M. Schwartze, Daniel Fiedler.

**Funding acquisition:** Anne C. Frenzel, Thomas Goetz, Corinna Reck.

**Investigation:** Manuel M. Schwartze.

**Methodology:** Manuel M. Schwartze, Daniel Fiedler.

**Project administration:** Manuel M. Schwartze, Anton K. G. Marx.

**Supervision:** Anne C. Frenzel.

**Writing – original draft:** Manuel M. Schwartze.

**Writing – review & editing:** Manuel M. Schwartze, Anne C. Frenzel, Thomas Goetz, Anton K. G. Marx, Reinhard Pekrun, Daniel Fiedler.

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
