## [Decision Letter · Decision Letter 0]

1 Sep 2020

PONE-D-20-20833

Excessive boredom among adolescents: A comparison between low and high achievers

PLOS ONE

Dear Dr. Goetz,

Thank you for submitting your manuscript to PLOS ONE. After careful consideration, we feel that it has merit but does not fully meet PLOS ONE’s publication criteria as it currently stands. Therefore, we invite you to submit a revised version of the manuscript that addresses the points raised during the review process.

We look forward to receiving your revised manuscript.

Kind regards,

Frantisek Sudzina

Academic Editor

PLOS ONE

Journal Requirements:

2. You indicated that you had ethical approval for your study. In your Methods section, please ensure you have also stated whether you obtained consent from parents or guardians of the minors included in the study or whether the research ethics committee or IRB specifically waived the need for their consent

Reviewers' comments:

Reviewer's Responses to Questions

**Comments to the Author**

1. Is the manuscript technically sound, and do the data support the conclusions?

Reviewer #1: Yes

Reviewer #2: Yes

Reviewer #3: Yes

2. Has the statistical analysis been performed appropriately and rigorously? 

Reviewer #1: Yes

Reviewer #2: Yes

Reviewer #3: Yes

3. Have the authors made all data underlying the findings in their manuscript fully available?

Reviewer #1: Yes

Reviewer #2: Yes

Reviewer #3: Yes

4. Is the manuscript presented in an intelligible fashion and written in standard English?

Reviewer #1: Yes

Reviewer #2: Yes

Reviewer #3: Yes

5. Review Comments to the Author

Reviewer #1: Review PONE-D-20-20833

Excessive boredom among adolescents: A comparison between low and high achievers

The authors ask an interesting and important question about boredom in high versus low achieving students. The sample and analysis procedures were adequate to answer the research question. The results provide further insight into students’ experiences of boredom as a negative and persistent emotion.

Page 1 “Are undesirable correlates of boredom augmented” – augmented might not be the right word choice. Maybe exacerbated? Or even just simply “worse”.

In the first paragraph, reduce the use of parentheses and add the i.e., likely over/under challenged to the main sentences because it is important content. Same in later parts of the manuscript. (thus likely over/under-challenged) should be part of the sentence.

The literature review is concise, to the point, and up-to-date. However, it would be helpful to add some more details on the “correlates” of boredom rather than using this as a general descriptor. In particular, what is the current evidence on boredom and behavioral and personality variables? This content should go before “The present study” section rather than having little bits of it in that section (e.g., references 23-35).

Please check the phrase “scholastic boredom” for accuracy. I think you just mean achievement boredom?

Only 258 of the original 1404 had high boredom? This seems quite small given the introduction puts the prevalence of boredom as 30-40%. This should be discussed maybe as a limitation or direction for future research differently than it is currently done because of its mismatch with current numbers. Also in the limitations section the n=271. Please check for accuracy between the two sections.

The discussion is appropriate and does not over generalize. However, it is a bit short. Can you please add some implications for the control-value theory especially because researchers in the area constantly have to manage the “boredom by over or under challenge” critique. I think this paper helps those researchers say that it doesn’t really matter – boredom is boredom.

Reviewer #2: In the study, pupils in fifth to tenth grades who were very bored and high-achieving in mathematics were compared with students who are very bored and low-achieving in mathematics. Comparisons were made regarding different behavioral and personality variables. Results show that there are no mean-level differences across the two groups for the investigated variables with three exceptions: Conduct problems and expressive suppression were higher for very bored low achievers and positive affect was higher for very bored high achievers.

The manuscript is well written; the research questions are interesting; and the methods and data analyses are adequate. Below I list some issues that could be addressed to improve the manuscript:

1) The use of terms and the categorization of the variables under consideration varies within the manuscript, sometimes considerably. In the abstract, the authors mention that behavior and personality aspects will be considered, but in the manuscript other super-categories are often used (e.g., on p. 8: emotional and behavioral problems, positive and negative affect, emotion regulation strategies, personality traits; on p. 13: behavioral and affective variables; Table 3: emotional symptoms). A standardization of categories and terms across the entire manuscript would strengthen the manuscript and help readers to follow the authors.

2) Moreover, the authors should consider providing a somewhat more detailed justification for the selection of the variables they consider.

3) A main problem of the study is the way in which the groups of high achievers and low achievers are defined. With a grading system from 1 to 6, it seems questionable whether students with average grades of 3 and 4 can actually be described as high achievers and low achievers respectively. It would be advantageous to exclude the group whose grade averages are in the middle of the applicable grading scale. It can be assumed that the students with 3s and 4s suffer less from boredom due to being overchallenged (i.e., the students with grade averages of a 4) or underchallenged (i.e., the students with grade averages of a 3) than students who actually perform particularly well or particularly poorly, especially since we know that classroom instruction is normally adapted to the needs of average-performing students. A comparison of the two ‘fringe groups’ (possibly using non-parametric tests) therefore seems a more germane approach to answer the question the authors are posing.

4) A discussion of the differences between school types would be important. Very bored low achievers in the low-achiever track of secondary school might differ from very bored low achievers in the high-achiever track of secondary school. Differences between the tracks could be analyzed or analyses could be controlled for school type.

5) At some points in the discussion the conclusions should be softened somewhat. For example, it cannot be clearly concluded from the results that boredom itself is associated with many problems regardless of scholastic performance, or that the situation seems to be even more dramatic for low-achieving students.

6) Minor issue: On page 10, the authors write that SDQ conduct and peer problems showed low reliabilities (alpha between .47 and .71). The second value does not match the value in Table 2 (and is not small).

Reviewer #3: Thank you for offering me the chance to review this paper. Overall, I liked the paper and I think it provides a new element to the research on academic boredom. The analyses are not complex but nonetheless adequate and done in an advanced way (e.g. latent construct correlations). It should be noted that the current study uses cross-sectional data only and that it relies on self-reported data as far as I could see.

General questions I had during reading the paper were:

While boredom was assessed specifically for the math class, the other measures seem to be of a more general nature. It seems like more general personal characteristics are thus inter-mixed with math-specific ones. How does this affect the conclusions that can be drawn?

Can there a stronger case be made regarding the degree to which the two selected groups are actually over/under challenged?

More specific questions/comments per section were:

Abstract

-Are you also referring to undesirable personality traits?

-Could replace ‘no study to date…’ with ‘merely knowing that student are highly bored’

Introduction

-the framing at the start of the paper could be geared somewhat more towards the Edu context.

-it is not entirely clear what ‘it’ refers to in the 5th sentence (…whether it is similarly…)

-I would suggest summarizing the theoretical and practical relevance explicitly and right away in the first para – ideally beyond ‘this has not been done before’… Also see ‘the present study’. Now the opening reads a bit anecdotical?

-p4 second para: I would like to propose to delete statements like ‘this has not been done before’ but rather explain what we can gain from investigating these things. Why is it important to know whether or not the same correlates exist for the two groups of students? Also, I do not see why conscientiousness would protect one from being bored? Conscientiousness could even strengthen feelings of being underchallenged? In sum, I think providing a stronger and more coherent rational for doing this investigation is needed.

Method

-the effective sample-size is 245 for the analysis of the main research question, this should be stated more explicitly, probably also in the abstract. There, only an N of 1404 students is mentioned describing the entire pool of students, which may raise wrong expectations. Alternatively, a more broad/general research question could be added?

-I wonder whether students of 9 year old can reliably answer questionnaires like the ERQ?

-Achievement: I suppose grades were self-reported? Please clarify in the paper.

-please explain how the chosen analyses can account for reliabilities low as .47/.53? It would probably have been more straight forward to leave these scales out of the analyses?

-please add the mean levels of boredom in the two groups that are created (or have I overlooked this?).

-How fair is it to consider these groups to be over-/under challenged? Is there something more that can be done to show this?

Analyses

-Wouldn’t it be more fitting to test interaction terms with low/high achievement and boredom?

-in any case: have you considered to correct the alpha levels for repeated testing?

Discussion

-I think the authors rightly conclude that it is in first place boredom as such rather than the combination with achievement that correlates with problematic student characteristics aso. Therefore, it may be a little overstated (especially given the very low R2s) that the situation is ‘more dramatic’ for low achievers.

-conduct problems had measurement issues, this should be mentioned along with the conclusions. Maybe it should receive a little less attention in the conclusions as well?

6. PLOS authors have the option to publish the peer review history of their article (what does this mean?). If published, this will include your full peer review and any attached files.

Reviewer #1: **Yes: **Lia M Daniels

Reviewer #2: No

Reviewer #3: **Yes: **Tim Mainhard

---

## [Editor Report · Decision Letter 1]

20 Oct 2020

Excessive boredom among adolescents: A comparison between low and high achievers

PONE-D-20-20833R1

Dear Dr. Goetz,

We’re pleased to inform you that your manuscript has been judged scientifically suitable for publication and will be formally accepted for publication once it meets all outstanding technical requirements.

Kind regards,

Frantisek Sudzina

Academic Editor

PLOS ONE

---

## [Editor Report · Acceptance letter]

28 Oct 2020

PONE-D-20-20833R1 

Excessive boredom among adolescents: A comparison between low and high achievers 

Dear Dr. Goetz:

I'm pleased to inform you that your manuscript has been deemed suitable for publication in PLOS ONE. Congratulations! Your manuscript is now with our production department. 

Kind regards, 

on behalf of

Dr. Frantisek Sudzina 

Academic Editor

PLOS ONE